# Reliability of Neural Implants—Effective Method for Cleaning and Surface Preparation of Ceramics

**DOI:** 10.3390/mi12020209

**Published:** 2021-02-19

**Authors:** Patrick Kiele, Jan Hergesell, Melanie Bühler, Tim Boretius, Gregg Suaning, Thomas Stieglitz

**Affiliations:** 1Laboratory for Biomedical Microtechnology, Department of Microsystems Engineering-IMTEK, University of Freiburg, 79110 Freiburg, Germany; Jan-Hergesell@web.de (J.H.); thomas.stieglitz@imtek.uni-freiburg.de (T.S.); 2Electrochemical Energy Systems, Department of Microsystems Engineering-IMTEK, University of Freiburg, 79110 Freiburg, Germany; Melanie.Buehler@imtek.uni-freiburg.de; 3Hahn-Schickard, 79110 Freiburg, Germany; 4Neuroloop GmbH, 79110 Freiburg, Germany; tboretius@neuroloop.de; 5School of Biomedical Engineering, The University of Sydney, Sydney 2006, Australia; gregg.suaning@sydney.edu.au; 6FRESCO Fellow, Freiburg Institute of Advanced Studies (FRIAS), University of Freiburg, 79104 Freiburg, Germany; 7Bernstein Center Freiburg, 79098 Freiburg, Germany; 8BrainLinks-BrainTools, 79110 Freiburg, Germany

**Keywords:** cleaning, Leslie’s soup, Teepol-L, isopropanol, deionised (DI) water, ceramic, contaminations, grease, flux, solder

## Abstract

Neural implants provide effective treatment and diagnosis options for diseases where pharmaceutical therapies are missing or ineffective. These active implantable medical devices (AIMDs) are designed to remain implanted and functional over decades. A key factor for achieving reliability and longevity are cleaning procedures used during manufacturing to prevent failures associated with contaminations. The Implantable Devices Group (IDG) at University College London (UCL) pioneered an approach which involved a cocktail of reagents described as “Leslie’s soup”. This process proved to be successful but no extensive evaluation of this method and the cocktail’s ingredients have been reported so far. Our study addressed this gap by a comprehensive analysis of the efficacy of this cleaning method. Surface analysis techniques complemented adhesion strengths methods to identify residues of contaminants like welding flux, solder residues or grease during typical assembly processes. Quantitative data prove the suitability of “Leslie’s soup” for cleaning of ceramic components during active implant assembly when residual ionic contaminations were removed by further treatment with isopropanol and deionised water. Solder and flux contaminations were removed without further mechanical cleaning. The adhesive strength of screen-printed metalisation layers increased from 12.50 ± 3.83 MPa without initial cleaning to 21.71 ± 1.85 MPa. We conclude that cleaning procedures during manufacturing of AIMDs, especially the understanding of applicability and limitations, is of central importance for their reliable and longevity.

## 1. Introduction

The applications of active implantable medical devices (AIMDs) increased enormously since the first fully implantable cardiac pacemaker in 1958 [1]. Nowadays, AIMDs take a central role in diagnosis and treatment of neurological disorders and diseases when pharmaceutical medicine comes to its limits. However, the challenge in the manufacturing of AIMDs is to fabricate long-term stable devices, which reliably fulfill their medical indication over decades, since the accessibility for maintenance is limited after implantation. Therefore, the understanding of individual manufacturing processes, their interactions among themselves and their impact on the longevity of the final device is crucial. One central aspect is the cleanliness of the device, which includes both cleaning and sterilisation of the final product to prevent infections in the surrounding tissue at the implant site as well as cleaning during individual manufacturing steps. While the first one prevents infections and inflammation by surface residues, the latter supports the adhesion of functional elements (e.g., metallisation layers) and thus ensures reliability and longevity of the implant. While the cleaning and sterilisation of implants has been reported widely in literature [2,3,4], there is a lack of applicable literature reporting cleaning procedures during the manufacturing of such a device. Nevertheless, the cleanliness in between sensitive assembly processes, in particular processes that have to provide adhesion between materials, is essential for reliable AIMDs. 

In the manufacturing of AIMDs, ceramics are often used due to their excellent chemical, mechanical and electrical properties (e.g., electrochemical stability, high mechanical strength and hermeticity, non-conductive, permeability for electromechanical waves for telemetric communication) while causing less imaging artefacts in magnetic resonance imaging (MRI) [5]. Typically, the ceramics serve as substrate material for printed circuit boards [6] or as hermetic housing [6,7]. A functionalisation of the ceramics is done by adding metal layers either in a screen-printing process or in a physical vapor deposition (PVD) process [7,8]. Later in the manufacturing chain, AIMDs are often casted in medical grade silicone serving as a non-hermetic encapsulation or protection of a hermetic housing from surrounding body fluid and vice versa [9] and increasing structural biocompatibility as a buffer between hard package and soft biological tissue material. However, the reliability and longevity of such implantable devices strongly depend on the adhesion of the functional layer and the encapsulation, the metallisation layer and the silicone cast, for example. In particular, the adhesion strongly depends on the initial cleanliness prior to processing. Furthermore, ionic contaminations are capable of attracting water via osmosis, leading to adhesion loss, degradation and failure of the implant [9].

To prevent failure due to insufficient adhesion of functional layers and ionic contamination on ceramics, a standardised cleaning procedure is required that is capable of removing organic and ionic contaminations. Many groups have reported the use of a cleaning procedure including “Leslie’s soup” and alcohol (e.g., isopropanol) followed by rinsing with deionised water. Herein, “Leslie’s soup” consists of the detergent Teepol-L (0.5 wt.%, Teepol Products, Kent, UK), Na_3_PO_4_·12 H_2_O (2.5 wt.%) and deionised water (97 wt.%). This procedure, or variations of it, has been applied to different applications ranging from implantable printed circuit boards (PCBs) with thick and thin film metallisations [10,11,12,13], electrode arrays [14] to non-hermetic [15,16] and hermetic packages [17] including electrical feed troughs [18,19]. Industry does not report on cleaning procedures, since it is their key intellectual property with respect to AIMD manufacturing.

Despite the wide application of the cleaning procedure for crucial manufacturing steps of AIMDs, there was surprisingly no specific study performed to evaluate quantitatively its cleaning performance.

According to personal communication, “Leslie’s soup” was named after and introduced by a British researcher of the aviation science to clean weld joints. From there, it found its way into the biomedical technology field for cleaning of ceramic components. 

In this study, we address the suitability of the “Leslie’s soup” cleaning procedure for the manufacturing of AIMDs. First, the impact of cleaning with “Leslie’s soup” on the adhesive strength of a screenprinted PtAu metallisation layer was shown. For a detailed investigation of the cleaning performance, common surface analysis techniques were applied [20]. In particular, we performed visual inspections and contact angle measurements followed by scanning electron microscopy and energy-dispersive X-ray spectroscopy to evaluate the cleaning performance. Furthermore, the ceramic samples under investigation were contaminated intentionally with burned flux, solder residues and grease to simulate harsh contaminations which can occur during the manufacturing of AIMDs.

## 2. Materials and Methods

### 2.1. Sample Preparation

The samples used in this work were 96% pure Al_2_O_3_ ceramic substrates (96% Rubalit 708S, CeramTec, Plochingen, Germany), which is commonly used in the manufacturing of AIMDs. For the evaluation of the cleaning procedure the substrates were contaminated with (1) grease, (2) flux and (3) solder residues simulating typical contaminations during the soldering process of AIMDs. Additionally, untreated—meaning “as clean as delivered”—substrates were evaluated. Grease (MPG50T, Electrolube, Leicestershire, UK) was applied uniformly on the substrate using a cotton swab. Flux (FL33E, EDSYN GmbH Europe, Kreuzwertheim, Germany) was applied uniformly followed by 20 s heating at 350 °C, to simulate the temperature impact during soldering. To contaminate samples with solder residues, two pieces (length: 1 cm; diameter: 0.5 mm) of solder (L-Sn60Pb38Cu2, EDSYN GmbH Europe, Kreuzwertheim, Germany) were placed on a ceramic substrate and heated to 350 °C with a solder iron. 

### 2.2. Cleaning Procedure

The cleaning procedure consisted of four consecutive steps. First, the samples were emerged in “Leslie’s soup” (0.5 wt.% Teepol-L, Teepol Products, Kent, UK, 2.5 wt.% Na_3_PO_4_·12H_2_O and 97 wt.% DI-water) and constantly swayed on a tumbling table for 5 min followed by swaying in isopropanol and DI-water for 5 min each. Lastly, the samples were cleaned in an ultrasonic bath in DI-water for additional 5 min. In this study, we did not perform any mechanical cleaning using brushes. In this way inaccessible or covered areas (e.g., by electronic components) were simulated, which may be present on circuit boards for AIMDs. 

### 2.3. Evaluation of Cleaning Effectiveness

The assessment of the effectiveness of the cleaning procedure was evaluated on performing contact angle measurements, SEM and EDX analyses. Further on, the adhesive strength of a screen-printed metallisation layer to the samples was determined by performing pull tests. 

The static contact angle of a water droplet on the particular sample was measured using the sessile drop method with a droplet size of 5 µL in an OCA20 Setup (Dataphysics GmbH, Filderstadt, Germany). All samples with different contaminations, as well as the “as clean as delivered” samples, were evaluated before and after the cleaning procedure. 

SEM and EDX analyses (Vega 3, Tescan, Germany, Dortmund, 10 kV, working distance of 15 mm) were performed on “as clean as delivered” samples only. By this, possible contaminations of the cleaning steps itself were evaluated. 

The adhesion strength of a metallisation layer on the ceramic substrate is of central importance for the long-term stability of AIMDs. Furthermore, it is highly dependent on the initial cleanliness of the substrate and can therefore be used as a measure for the cleanliness of the substrate itself. Two PtAu solder pads (2 mm × 2 mm) were screen-printed on Al_2_O_3_ substrates which were not cleaned with the described cleaning procedure. For a comparison of the adhesive strength to the recommendations of the safety limits stated by material supp-liers of glass-pastes [21], we performed an axially loaded butt joint test. For this, wires were soldered manually to the PtAu pads to characterise the adhesive strength of the layer stack in a pull testing setup (Instron Corporation, Norwood, MA, USA).

### 2.4. Statistical Analysis

A one-way ANOVA analyses was used to determine the effectiveness of the cleaning procedure including all different contaminations (Cleaning was not on the same sample). A multicomparision of all groups (contaminated and cleaned) was performed using Fisher test. The significance was set at the α level of 0.05. All statistical analysis in this study were performed using OriginPro 2017G (OriginLab Corporation, Northampton, MA, USA).

## 3. Results

The adhesion of a screen-printed PtAu metallisation layer on Al_2_O_3_ substrates was dependent on the condition of the substrate. The metallisation layer without cleaning of the substrate prior to the screen-printing process resulted in an adhesive strength of 12.50 ± 3.83 MPa. When applying the cleaning procedure to the ceramic substrate the adhesive strength exceeded the safety threshold of 17 MPa for reliable pad adhesion (21.71 ± 1.85 MPa) (Figure 1). The breakages were of cohesive nature and occurred in all cases at the ceramic-metallisation interface.

Ceramic substrates “as clean as delivered”, as well as with different initial contaminations (solder, grease and flux), were subject to the full cleaning procedure including shaking in “Leslie’s soup”, isopropanol and DI-water finalised by an ultrasonic treatment in DI-water. The substrates featured no visible contaminations after cleaning (Figure 2).

### 3.1. Contact Angle

The contact angle measurements showed wide variations for the specific contaminations (Figure 3). The “as clean as delivered” substrate featured a contact angle of 41.0 ± 5.6°. As expected, the applied contaminations changed the wettability of the substrate resulting in contact angles of 68.7 ± 14.5° for the solder contamination and 96.0 ± 4.3° for the grease contamination indicating hydrophobic surfaces. The flux contamination decreased the contact angle to 12.7 ± 5.2°. 

After performing the full cleaning procedure the contact angle of the initially “as clean as delivered”, solder and flux contaminated substrates was 48.4 ± 10.6°, 50.5 ± 7.8° and 47.8 ± 58° respectively. The contact angle of the initially grease contaminated substrate was higher (61.0 ± 5.9°) after cleaning compared to the other groups. There were significant variations comparing the contact angles after cleaning of the different contaminated samples (F = 75.02, *p* < 0.001; ANOVAoneWay). Specifically, there were significant changes in the contaminations Solder, Grease and Flux (each *p* < 0.01; Fisher test), whereas cleaning had no significant effect on the contact angles of the "as clean as delivered" sample (p_w/o-w/cleaning, as delivered_ = 0.093; Fisher test). Despite the significant differences between the contaminations before cleaning (each *p* < 0.001; Fisher test), the multicomparison showed that there were no significant differences between the “as clean as delivered”, solder and flux left after treating the samples with the cleaning procedure. However, the contact angle of grease after cleaning was significantly higher compared to “as clean as delivered” (*p*_as delivered-grease_ = 0.011), solder (*p*_solder-grease_ = 0.031) and flux contaminations after cleaning (*p*_flux-grease_ = 0.007; Fisher test). 

The significance of an initial cleaning step with “Leslie’s soup” was demonstrated on substrates contaminated with grease and flux. For this purpose, the first step "swaying in “Leslie’s soup” for 5 minutes" was omitted when cleaning the substrates. Accordingly, after cleaning only with isopropanol and deionised water, the contact angles were 76.1 ± 9.1° (n = 9) for flux contaminants and 103.5 ± 6.4° (n = 9) for grease contaminants. The same contaminations were removed with “Leslie’s soup” only, neglecting the subsequent cleaning steps. Here, the contact angles were in both cases smaller than the detection limit of 10° (n = 9 each).

### 3.2. EDX Analysis

The surface elemental analyses after the single cleaning steps of samples “as clean as delivered” indicated an increased content of sodium (10%) and phosphor atoms (3%) after the treatment with “Leslie’s soup”. After performing the full cleaning procedure these residues dropped to 1%. Initially, a carbon concentration of 2% was present, which dropped to 1% after the treatment with “Leslie’s soup” and isopropanol (Table 1). The “as clen as delivered” sample revealed some inhomogeneities on the surface at a magnification of 100, which were not present after performing the full cleaning procedure (Figure 4).

## 4. Discussion

In the manufacturing of AIMDs, adhesion of different materials plays a major role for device reliability over decades and is therefore of central importance in a variety of assembly and packaging techniques. Often, metallisation layers on ceramic substrates are used for the implementation of PCBs or for solder joints in hermetic housing concepts. Silicone casts are used either as a non-hermetic housing, or to protect hermetic housing from corrosion and mechanical loads. In all applications the adhesion of the involved materials is crucial to prevent delamination and failure [9]. To ensure good adhesion, the substrates must be cleaned sufficiently. 

In this study, the adhesion or the adhesive strength of metal layers on ceramic substrates was used as a measure of the substrate’s cleanliness to demonstrate the impact of an only slightly contaminated substrate. The adhesion of screen-printed metal PtAu metal structures was strongly dependent on the condition of the substrates. Without cleaning of the ceramics, the adhesive strength was below the manufacturer’s safety limit (17 MPa) [21] resulting possibly in a failure of the entire implant, even though these ceramics are usually supplied in a sufficiently clean condition, provided that the surface is not touched and kept free of dust [18,22]. By performing the proposed cleaning procedure prior to screen-printing, the adhesive strength exceeded the limit, indicating a successful cleaning of the substrate. Thus, some form of cleaning is absolutely necessary, even if the substrates are brand new. After the necessity of cleaning has been demonstrated, typical boundary conditions in the manufacturing of medical devices were defined and the present cleaning method was evaluated in this respect.

In addition to the manufacturing of screen-printed PCBs for AIMDs, the assembly of electronic components, in particular the soldering and the removal of associated contaminations, might be crucial for following manufacturing steps. Often, the contaminations are difficult to reach when using mechanical cleaning methods. Therefore, the applied cleaning procedure should not rely on manual mechanical cleaning steps. Instead of performing a detailed analysis based on adhesion forces reflecting only a specific material combination and their process parameters (e.g., solvents in screen-printing paste and firing temperature), we used standard surface analyses methods [20], namely visible inspection, contact angle measurements, SEM and EDX analyses to evaluate the cleaning performance instead.

The visible inspection of the cleaned substrates with initial contaminations originating from assembly steps indicated a successful cleaning performance when using the present procedure including “Leslie’s soup”, isopropanol, DI-water and an ultrasonic treatment. However, to gain more insights in the cleaning performance, we performed contact angle measurements before and after cleaning of contaminated substrates and a step-by-step surface analysis from the initial towards the cleaned samples. 

### 4.1. Contact Angle Measurements

Due to the sensitivity of the wetting of solid surfaces to contaminations the contact angle can be used as a measure for the cleanliness [23]. Solder and flux residues were hydrophobic resulting in higher contact angles than the “as clean as delivered” ceramic, while it was the opposite for flux contaminations. However, having both, hydrophilic and hydrophobic contamination led us to the assumption that the cleaning performance was successful whenever the resulting contact angles of all samples after cleaning were at the same level. This was true for the solder and flux contaminated samples as well as for the “as clean as delivered” substrate. The contact angle of the grease contaminated samples decreased drastically after cleaning but remained significantly higher than the contact angles of the other substrates after cleaning. Based on these measurements, greasy samples require further cleaning, e.g., mechanical cleaning. We therefore highly recommend eliminating sources of grease during the manufacturing process, even if, the initial level of the purposely applied contamination in this study was rather high compared to the more realistic case during the assembly of implants. The other, more typical contaminations during the assembly of AIMDs (solder, flux) were removed completely based on the comparison of the contact angles. Nevertheless, this method does not prove whether there are contaminations left after cleaning, appearing from the cleaning steps itself, which could result in similar contact angles. 

### 4.2. Step-by-Step Evaluation

The success of each individual cleaning step was evaluated based on EDX analyses of an “as clean as delivered” substrate. After the treatment with “Leslie’s soup” the concentration of sodium and phosphate on the surface increased, while the concentration of carbon did not change. However, the tensides in “Leslie’s soup” or Teepol-L alter the binding forces between contamination and the substrate which leads to a detachment. Additionally, Na_3_PO_4_ forms an alkaline environment, supporting the cleaning effect of the tensides [24]. This cleaning effect can be seen in the resulting contact angles when using “Leslie’s soup” alone compared to cleaning with isopropanol and deionised water. Isopropanol and deionised water had no major influence on the contact angle indicating no cleaning effect of the contaminations. The very low contact angle after treating with “Leslie’s soup” indicates a removal of flux and grease resulting in a highly hydrophilic surface. Accordingly, further cleaning steps are necessary. Therefore, the following treatment with isopropanol was included to remove remaining organic contaminations [25], which can be seen in the small drop of the carbon concentration. To remove ionic residues from the earlier cleaning steps and, if present, from initial contaminations, the samples were shaken in DI-water [25]. The cleaning effect of DI-water was further increased by applying an ultrasonic treatment. After performing all cleaning steps, only aluminium and oxide, originating of the ceramics itself were observed. The residual contents of C, Na and P did not manifest a specific peak in the EDX spectrum and were allocated to noise in the EDX spectra. 

The SEM images comparing the initial with the cleaned surface of an “as clean as delivered” substrate showed evidence of the cleaning effect with the full treatment since the initial irregularities on the ceramic surface vanished.

### 4.3. Limitations of the Study

In this study, we demonstrated the performance of the cleaning procedure with “Leslie’s soup”, isopropanol, DI-water and an ultrasonic treatment with three types of contaminations on flat Al_2_O_3_ substrates. However, the cleaning effect might be different with other materials, contaminations or geometries. Even though we showed the successful removal of flux, solder and partly grease contaminations in this study, it is recommended to validate the cleaning performance individually on the specific framework. In this study, we did not use mechanical cleaning methods to simulate the cleaning of hard-to-reach areas. Nevertheless, it is recommended that the design allows an initial mechanical cleaning (e.g., brushing). This reduces initial contamination and thus increases the cleaning success of tough contaminants (e.g., grease). Furthermore, a final rinsing step with deionised water while monitoring the ionic conductivity is also advisable to gather continuously quantifiable data for a process control [26].

The presented cleaning procedure is intended to clean components during individual manufacturing steps to increase the adhesive integrity. An additional encapsulation of the product with a polymer (e.g., silicone rubber) is common to improve structural biocompatibility. The presented cleaning procedure does not replace final cleaning and sterilisation of the product, which prevents infections and inflammation caused by surface residues. For this purpose, we recommend the use of standardised cleaning and sterilisation procedures [2,3,4]. 

## 5. Conclusions

We demonstrated the performance of a cleaning procedure including “Leslie’s soup”, which is suitable for removing solder, flux and partly grease contaminations on ceramic substrates without manual mechanical treatment. The cleaning procedure contains the consecutive processing steps: swaying in “Leslie’s soup”—a mixture of the detergent Teepol-L, Na_3_PO_4_∙12 H_2_O and deionised water—isopropanol and deionised water followed by an ultrasonic treatment. Although it does not replace final cleaning and sterilisation of the product, this cleaning process for intermediate steps in the manufacturing process provides another step for safe implants. Disclosure and quantitative assessment of important steps in active implantable medical device manufacturing allows better comparability in neural implant development and contributes to higher reliability and longevity in translational research and transfer of research results into medical products.

## Figures and Tables

**Figure 1 micromachines-12-00209-f001:**
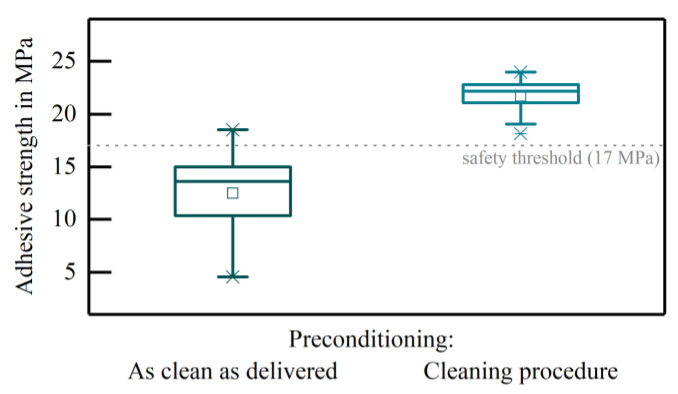
Adhesive strength of PtAu metallisation layers on Al_2_O_3_ substrates without (n = 17) and with a treatment with the cleaning procedure (n = 10) prior to screen-printing.

**Figure 2 micromachines-12-00209-f002:**
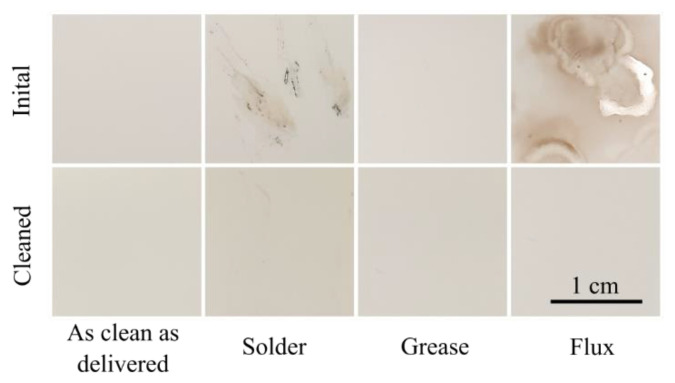
Visual comparison of the ceramic substrates before and after performing the cleaning procedure. The procedure was performed on “as clean as delivered” samples as well as on samples with solder, grease and flux contaminations.

**Figure 3 micromachines-12-00209-f003:**
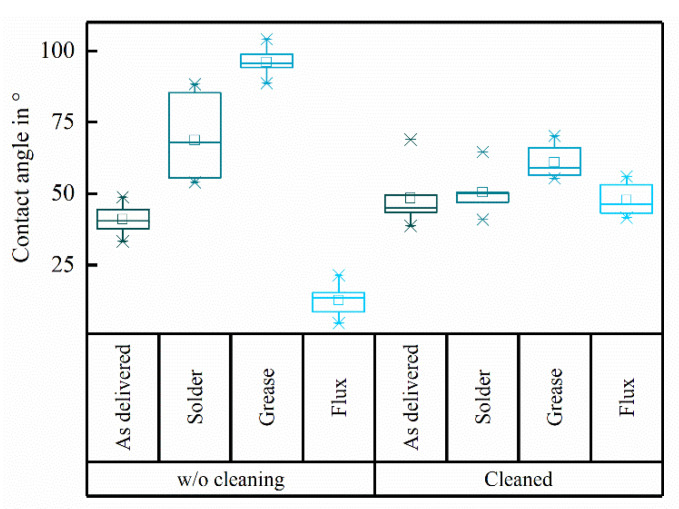
Contact angles of different contaminations before (n = 9) and after (n = 6) cleaning in “Leslie’s soup”, isopropanol and DI-water followed by an ultrasonic treatment.

**Figure 4 micromachines-12-00209-f004:**
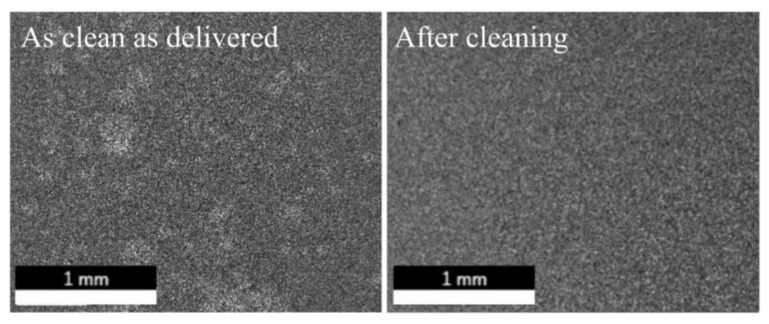
Comparison of the surfaces of an “as clean as delivered” ceramic substrate and after performing the cleaning procedure.

**Table 1 micromachines-12-00209-t001:** Elemental analysis of the ceramic surfaces after the single process steps of the cleaning procedure.

Treatment	C in %	O in %	Na in %	Al in %	P in %
As clean as delivered	2	35	1	65	1
Leslie’s soup	2	38	10	46	3
+ Isopropanol	1	36	13	46	4
+ DI-water	1	33	2	61	2
+ ultrasonic	1	38	1	59	1

## Data Availability

The data presented in this study are available on request from the corresponding author.

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
