# Peer review of "Reliability of Neural Implants—Effective Method for Cleaning and Surface Preparation of Ceramics"

_micromachines, 2021, doi:10.3390/mi12020209_

Round 1

Reviewer 1 Report

The authors performed a quantitative study to assess the suitability of “Leslie’s soup” for surface cleaning of ceramic components in neural implants. While the procedures and evaluation methods were clearly described, an in vitro biocompatibility test (cell culture) is recommended given that 1) the surface and tissue interaction is unexplored in this study and 2) surface cleanliness and cell interactions can impact the longevity of the implantable devices. 

A minor comment: please check spelling mistakes. For example, line 76 "a standardized cleaning procedure is required wich is capable"

Author Response

Reviewer 1: (R1)
Author: (A)

R1: The authors performed a quantitative study to assess the suitability of “Leslie’s soup” for surface cleaning of ceramic components in neural implants.

A: Thank you very much.

R1: While the procedures and evaluation methods were clearly described, an in vitro biocompatibility test (cell culture) is recommended given that 1) the surface and tissue interaction is unexplored in this study and 2) surface cleanliness and cell interactions can impact the longevity of the implantable devices. 

A: We appreciate this valid criticism. However, this study is designed to show the necessity and procedure of cleaning in between individual manufacturing steps to increase the longevity of such devices. The ceramic always will be coated with an additional polymer as encapsulation before implantation. The described cleansing should not replace a final cleaning and sterilization of the product according to the applicable standard. We mentioned this already in our introduction (“One central aspect is the cleanliness of the device, which includes both cleaning and sterilization of the final product to prevent infections in the surrounding tissue at the implant site as well as cleaning during individual manufacturing steps. While the first one prevents infections and inflammation by surface residues, the latter supports the adhesion of functional elements (e.g. metallization layers) and thus ensures reliability and longivity of the implant.”) and conclusion (“The cleaning procedure contains the consecutive processing steps: swaying in “Leslie’s soup” - a mixture of the detergent Teepol-L, Na3PO4∙12 H2O and deionized water - isopropanol and deionized water followed by a ultrasonic treatment. Although it does not replace final cleaning and sterilization of the product, this cleaning process for intermediate steps in the manufacturing process provides another step for safe implants.”).

Despite the information we have already given, it is extremely important to clearly indicate the additional need for cleaning and sterilization of the final product. Accordingly, we have added the following additional sentences in the section “4.3 Limitations of the study”: “The presented cleaning procedure is intended to clean components during individual manufacturing steps to increase the adhesive integrity. An additional encapsulation of the product with a polymer (e..g. silicone rubber) is common to improve structural biocompatibilty. The presented cleaning procedure does not replace final cleaning and sterilization of the product, which prevents infections and inflammation caused by surface residues. For this purpose, we recommend the use of standardized cleaning and sterilization procedures [2–4].” (Line: 311-317)

R1: A minor comment: please check spelling mistakes. For example, line 76 "a standardized cleaning procedure is required wich is capable"

A: Thank you for this comment, we have done an extensive grammar and spelling check.

All line numbers refer to the revised manuscript.

Reviewer 2 Report

The authors investigated the performance of “Leslie’s soup” on ceramics under different conditions: as new as delivered, solder contaminated, grease contaminated, and flux contaminated. While the work can be useful in manufacturing, several key components are missing.

  1. Figure 1. The binding strength is only compared between ‘as new as delivered’ and after washes. What the binding of contaminated ceramics before and after washing? Also, it is unexpected that almost none of the ‘as clean as delivered’ ceramics meet the safety threshold in bonding strength. Does that mean washing is a mandatory step before plating metal, even if the ceramic is brand new?
  2. The authors did not explain how the adhesion tests were conducted. Do the tests simulate the actual attachment of the metallization layer?
  3. Figure 3 shows that there is grease residue on the ceramic. The authors should figure out a way to improve the protocol for grease removal and include the updated protocol.
  4. From Table 3, it seems that Leslie’s soup is not quite efficient in removing organic materials based on the carbon change before and after the soup washing. Also, it seems that isopropanol outperforms Leslie’s soup in removing organic materials.
  5. To prove the necessity and efficiency of Leslie’s soup, the authors should wash the ceramic with the protocol without Leslie’s soup and check the outcome like adhesion strength, contact angle, and residue components.

Author Response

Reviewer 2:  (R2)

Author: (A)

R2: The authors investigated the performance of “Leslie’s soup” on ceramics under different conditions: as new as delivered, solder contaminated, grease contaminated, and flux contaminated. While the work can be useful in manufacturing, several key components are missing.

A: Thank you very much for this assessment.  

R2: 1) Figure 1. The binding strength is only compared between ‘as new as delivered’ and after washes. What the binding of contaminated ceramics before and after washing? Also, it is unexpected that almost none of the ‘as clean as delivered’ ceramics meet the safety threshold in bonding strength. Does that mean washing is a mandatory step before plating metal, even if the ceramic is brand new?

A: We appreciate this valid criticism. In fact, it is true that almost none "as cleaned as delivered" reach the safety limit. Unfortunately, this is a fact unknown to many. Based on the adhesion tests or Figure 1, we wanted to show the essential necessity of cleaning, even if the substrates are brand new. We commented on this in the discussion: “Without cleaning of the ceramics, the adhesive strength was below the manufacturer’s safety limit (17 MPa) [22] resulting possibly in a failure of the entire implant, even though these ceramics are usually supplied in a sufficiently clean condition, provided that the surface is not touched and kept free of dust [18,23].”

Additional adhesion tests with initially contaminated substrates would probably detract from this statement. Furthermore, it is by no means new that, especially greasy contamination reduces or prevents adhesion. Accordingly, an extension of the adhesion tests would not add any scientific value.

We clarified this statement by adding the following sentence: “Thus, some form of cleaning is absolutely necessary, even if the substrates are brand new.” (Line: 238-239)

R2: 2) The authors did not explain how the adhesion tests were conducted. Do the tests simulate the actual attachment of the metallization layer?

A: The method we used for adhesion testing is described in “2.3 Evaluation of cleaning effectiveness“: (“Two PtAu solder pads (2 x 2 mm²) were screen-printed on Al2O3 sub­strates which were not cleaned with the described cleaning procedure. For a comparison of the adhesive strength to the recommendations of the safety limits stated by material supp­liers of glass-pastes [22], we performed an axially loaded butt joint test. For this, wires were soldered manually to the PtAu pads to characterize the adhesive strength of the layer stack in a pull testing setup (Instron Corporation, Norwood, MA, USA).”). Nevertheless, we must admit that we have not further described the nature of the failure. We added the following statement: “The breakages were of cohesive nature and occurred in all cases at the ceramic-metallization interface.” (Line: 159-160)

R2: 3) Figure 3 shows that there is grease residue on the ceramic. The authors should figure out a way to improve the protocol for grease removal and include the updated protocol.

A: Many thanks for this valuable comment. We added this information in the revised manuscript in the “4.3: Limitations of the study” section: “In this study, we did not use mechanical cleaning methods to simulate the cleaning of hard-to-reach areas. Nevertheless, it is recommended that the design allows an initial mechanical cleaning (e.g. brushing). This reduces initial contamination and thus increases the cleaning success of tough contaminants (e.g. grease).” (Line: 305-308)

R2: 4) From Table 3, it seems that Leslie’s soup is not quite efficient in removing organic materials based on the carbon change before and after the soup washing. Also, it seems that isopropanol outperforms Leslie’s soup in removing organic materials.

A: Thank you for this point of criticism. Our comments on 4) can be found in the response to 5).

R2: 5) To prove the necessity and efficiency of Leslie’s soup, the authors should wash the ceramic with the protocol without Leslie’s soup and check the outcome like adhesion strength, contact angle, and residue components.

A: We agree with the reviewer, that table 3 is misleading regarding the necessity of Leslie’s soup. Moreover, we have not reported a comparison to cleaning without Leslie's soup. Unfortunately, we must admit that we missed to include this in the original manuscript. We added an evaluation of the resulting contact angle after cleaning grease and flux contaminations with isopropanol and deionized water only as well as with Leslie’s soup only in the results section: “The significance of an initial cleaning step with Leslie's soup was demonstrated on substrates contaminated with grease and flux. For this purpose, the first step "swaying in Leslie's soup for 5 minutes" was omitted when cleaning the substrates. Accordingly, after cleaning only with isopropanol and deionized water, the contact angles were 76.1 ± 9.1° (n=9) for flux contaminants and 103.5 ± 6.4° (n=9) for grease contaminants. The same contaminations were removed with Lesie’s soup only, neglecting the subsequent cleaning steps. Here, the contact angles were in both cases smaler than the dection limit of 10° (n = 9 each).” (Line: 189-205)

These results are discussed in the section “4.2 Step by Step evaluation”: “This cleaning effect can be seen in the resulting contact angles when using Lesie’s soup alone compared to cleaning with isopropanol and deionized water. Isopropanol and deionized warter had no major influence on the contact angle indicating no cleaning effect of the contaminations. The very low contact angle after treating with Leslie’s soup indicates a removal of flux and grease resulting in a highly hydrophilic surface. Accordingly, further cleaning steps are necessary.“ (Line 281-286)

Additional comment: We have done an extensive grammar and spelling check. All line numbers refer to the revised manuscript.

Round 2

Reviewer 1 Report

No further questions. 

Author Response

Reviewer 1: (R1)
Author: (A)

R1: No further questions. 

A: Thank you very much.

Reviewer 2 Report

I'm not convinced by the author's statement that adhesion tests are not necessary for contaminated materials after cleaning with Leslie's soup. As discussed in the manuscript (Line 99), different contamination conditions were introduced to simulate the harsh contaminations that can occur during the manufacturing of AIMDS. The authors did multiple test, such as visual inspection, contact angle change, to prove that the use of Leslie's soup largely removes these contaminations. And the authors claim through that the adhesion integrity of the ceramic substrate is of importance for the longevity of AIMDS. I think it is more straightforward to inspect the performance of Leslie's soup by comparing the adhesion strength change of contaminated ceramic substrate before and after Leslie's soup cleaning.

Author Response

Reviewer 2:  (R2)

Author: (A)

R2: I'm not convinced by the author's statement that adhesion tests are not necessary for contaminated materials after cleaning with Leslie's soup. As discussed in the manuscript (Line 99), different contamination conditions were introduced to simulate the harsh contaminations that can occur during the manufacturing of AIMDS. The authors did multiple test, such as visual inspection, contact angle change, to prove that the use of Leslie's soup largely removes these contaminations. And the authors claim through that the adhesion integrity of the ceramic substrate is of importance for the longevity of AIMDS. I think it is more straightforward to inspect the performance of Leslie's soup by comparing the adhesion strength change of contaminated ceramic substrate before and after Leslie's soup cleaning.

A: Thank you very much for sharing your strong opinion.

In this manuscript, we generally want to address the high need for cleaning steps and a methodology for doing so. In this case, the adhesion test serves as one example to highlight the need for cleaning, even if the substrates are brand new. We agree with the reviewer that it is more straightforward to demonstrate the cleaning success of the substrates using further adhesion tests. However, in this case, only a special case, namely the material combination of PtAu screen printing paste on alumina, would be shown. For example, the temperature profile during firing or solvents in the screen printing paste could distort the result for other material combinations (e.g. sputtered thin film metallization on alumina [1] or silicone encapsulations on alumina [2]). To address this factor, we decided to use established surface analysis methods [3] instead of pull-test for evaluation of the adhesion strength. These analysis methods include optical inspection, contact angle measurements, scanning electron microscopy analyses, and electron spectroscopy for chemical analyses, among others. We believe that our chosen analytical methods have resulted in a more robust and comparable study that will be of higher interest to a wider readership of the journal. A comparative study on the outcome of these cleaning steps on different thick-film and thin-film materials in combination with additional soldering and bonding processes would complete the big picture. This work, however, would go far beyond the scope and focus of our submitted manuscript.

To clarify this in the manuscript, we added following statements:

 Line 230-231: “In this study, the adhesion or the adhesive strength of metal layers on ceramic substrates was used as a measure of the substrate’s cleaniness to demonstrate the impact of an only slightly contaminated substrate.”

And (Line: 247-250): “Instead of performing a detailed analyses based on adhesion forces reflecting only a secific material combination and their process parameters (e.g. solvents in screen-printing paste and firing temperature), we used standard surface analyses methods [21], namely visible inspection, contact angle measurements, SEM and EDX analyses to evaluate the cleaning performance instead.”

References

[1]   Kiele P, Cvancara P, Langenmair M, Mueller M and Stieglitz T 2020 Thin Film Metallization Stacks Serve as Reliable Conductors on Ceramic-Based Substrates for Active Implants IEEE Trans. Compon., Packag. Manufact. Technol. 10 1803–13

[2]   Vanhoestenberghe A and Donaldson N 2013 Corrosion of silicon integrated circuits and lifetime predictions in implantable electronic devices J. Neural Eng. 10 31002

[3]   Critchlow G W and Brewis D M 1995 Review of surface pretreatments for titanium alloys International Journal of Adhesion and Adhesives 15 161–72